# Fisher Divergence for Attribution through Stochastic Differential Equations

## Abstract

Deep neural networks achieve remarkable performance but often lack interpretability, raising concerns in critical applications. Feature attribution methods, including perturbation-based methods, aim to address this by quantifying the contribution of input features to model outputs. However, existing methods often rely on narrowly defined perturbation spaces or sampling within the predefined large perturbation space, leading to incomplete or misleading explanations, especially in high-dimensional settings. To overcome these limitations, we propose a novel perturbation-based framework to leverage Stochastic Differential Equations to model continuous perturbations and comprehensively explore the input space in an effective way. By connecting Fisher Divergence with the time derivatives of KL divergence and mutual information, our approach provides a rigorous theoretical foundation for quantifying feature importance. Additionally, we integrate the Information Bottleneck (IB) principle into an optimization framework, ensuring the identification of the most informative features while maintaining predictive performance.

## 1. Introduction

Deep neural networks (DNNs) have demonstrated remarkable success in a wide range of applications, including computer vision, and natural language processing. Despite their impressive performance, DNNs often function as "black boxes," providing limited insight into their underlying decision-making processes (Pan et al., 2021; Novello et al., 2022a; Chen et al., 2024). This lack of transparency raises concerns in high-stakes domains where understanding the rationale behind model predictions is vital for trust, accountability, and adherence to regulatory guidelines (Chaddad et al., 2023; Saraswat et al., 2022; Soundararajan & Shenbagaraman, 2024). Consequently, explainability has emerged as a pivotal research area, aiming to elucidate how neural networks transform inputs into outputs (Van der Velden et al., 2022; Bai et al., 2021).

One well-established strategy for enhancing neural network explainability is feature attribution, which quantifies how each input feature contributes to a model's output (Zhou et al., 2022). Although various techniques exist to achieve this, perturbation-based methods—which systematically modify or remove parts of the input and then measure changes in the model's predictions—are particularly intuitive, offering a clear and direct interpretation of how the model reacts to input modifications and their influence on its behavior (Ivanovs et al., 2021).

A straightforward perturbation-based approach is to consider a feature as important if a small changes leads to a significant prediction change (**Small Perturbations → Large Output Changes**). Gradient-based methods operationalize this idea by computing the gradient of the output with respect to the input (Simonyan et al., 2014; Sundararajan et al., 2017; Smilkov et al., 2017). However, a major limitation of methods relying on small input variations is their dependence on narrowly defined perturbation spaces (Fel et al., 2023), which may fail to trigger changes in the function's output, thereby overlooking certain aspects of feature importance. For example, in the case of piecewise functions, a change in input must reach a certain threshold to activate an output shift, which small perturbations are unlikely to achieve. Consequently, larger-scale perturbations could offer a more comprehensive perspective on feature importance, capturing a wider range of real-world variations.

When the perturbation space is broadened to allow larger input changes, both inputs and outputs can shift significantly (**Large Perturbations → Large Output Changes**). In such scenarios, to assess feature importance is to linearly or non-linearly distribute the total output change across individual input features based on its share of the output change. However, linear decomposition is only suitable for small-scale perturbations. Methods like DeepLIFT (Shrikumar et al., 2017) extending it to larger input changes lack a solid theoretical foundation and may lead to inaccurate attributions. Non-linear decomposition methods like SHAP and its variants (Lundberg & Lee, 2017; Jethani et al., 2021; Tsai et al., 2023), while theoretically sound, are computationally expensive.

Another perspective identifies features whose substantial alteration causes minimal impact on the model's output,

thereby highlighting important features in the input (**Large Perturbations → Small Output Changes**). However, exploring large input changes demands navigating a vast perturbation space, which is computationally expensive, especially for high-dimensional data like images. To mitigate this, many methods reduce the complexity by partitioning the inputs into smaller blocks (Novello et al., 2022a) or by sampling within the predefined perturbation space (Fel et al., 2023) rather than performing an exhaustive search. Although these techniques lower computational overhead, they also sacrifice precision: coarse partitioning overlooks fine-grained details, and limited sampling may fail to represent the entire perturbation space, potentially leading to imprecise or misleading attributions.

To address these limitations, we propose a novel perturbation-based explainability method that leverages Stochastic Differential Equations (SDEs). By modeling the perturbation process as an SDE, we introduce a continuous-time stochastic process that perturbs the data distribution with continuously varying levels of noise. This approach allows us to explore the input space more comprehensively, capturing richer information about feature importance and interactions. To mitigate computational complexity, we construct an optimization framework that identifies the distribution of the most significant features, avoiding an exhaustive search of the entire perturbation space.

Central to our optimization framework is the introduction of Fisher Divergence in terms of the score function, defined as the derivative of the distribution of the most important features with respect to the input features. And the time integration of Fisher Divergence serves as a quantitative measure of feature importance, reflecting how changes in input features influence the distribution of important features identified by our method.

Our contributions can be summarized as follows:

- **SDE for Perturbations Space:** We employ SDEs to define an unconstrained perturbation space, rather than a predefined space of fixed size.

- **An Optimization Framework:** We propose an optimization framework for feature attribution in an unconstrained perturbation space, avoiding the computational burden of exhaustively searching the entire space.

- **Linking Between Mutual Information and Fisher Divergence:** By linking mutual information with Fisher divergence, we provide a principled information-theoretic perspective for quantifying feature importance, enriching the theoretical framework of perturbation-based explainability.

- **Connection Between KL Divergence and Fisher Divergence:** We establish a novel theoretical relationship

by deriving the time derivative of KL divergence and linking it to Fisher divergence. Since mutual information can be formulated in terms of KL divergence, this connection offers a principled approach to computing mutual information.

## 2. Related Works

Explainable methods based on input perturbations can be broadly categorized according to how changes in inputs (small or large) correlate with changes in the model's output (small or large). Below, we discuss three main approaches: (1) small perturbations leading to large output changes, (2) large perturbations leading to large output changes, and (3) large perturbations leading to small output changes.

**Small Perturbations → Large Output Changes:** If a minor change in a particular feature leads to a substantial difference in the model's output, that feature is deemed influential. Gradient-based methods adopted this strategy via gradients. Saliency Maps (Simonyan et al., 2014) compute these gradients to highlight important features. Integrated Gradients (Sundararajan et al., 2017) and SmoothGrad (Smilkov et al., 2017) also utilize this concept but employ different data augmentation strategies to enhance the estimation of feature importance. Grad-CAM (Selvaraju et al., 2017a) preserves model flexibility by leveraging the gradients of the target class with respect to the activations in the last convolutional layer, highlighting regions of importance. Building on this, Guided-GradCAM (Selvaraju et al., 2017b) enhances Grad-CAM by integrating it with Guided Backpropagation (Springenberg et al., 2014), providing finer and more detailed insights into feature importance.

**Large Perturbations → Large Output Changes:** Various methods employ this principle of "big changes, big effects." RISE (Petsiuk et al., 2018) uses multiple random binary masks to occlude large portions of the input; the change in model output across many masked samples is aggregated into a saliency map. LIME (Ribeiro et al., 2016) perturbs different subsets of features and then fits a simple local model to approximate how drastically these features affect the original prediction. Meanwhile, SHAP (Lundberg & Lee, 2017) and its variants (FaithSHAP, FastSHAP (Jethani et al., 2021; Tsai et al., 2023)) draw on the game-theoretic concept of Shapley values to systematically credit (or blame) each feature for the overall output change when large combinations of features are removed. Beyond single-pass masking, Remove and Retrain (ROAR) (Hooker et al., 2019) re-trains the model after removing key features, measuring performance deterioration to verify feature importance. Lastly, Explaining by Removing (Covert et al., 2021) provides a unified theoretical framework for such removal-based strategies, analyzing how various large-perturbation methods align with desired explanation properties.

**Large Perturbations → Small Output Changes:** This approach focuses on identifying features that are critical to the model's prediction even when large portions of the input are altered but the output change remains minimal. For instance, studies have shown that improving network interpretability can directly enhance adversarial robustness by ensuring predictions rely on stable feature subsets (Boopathy et al., 2020). Abduction-based explanations have been proposed to identify invariant features that maintain predictions under significant input perturbations (Ignatiev et al., 2019), while metrics for representativity and consistency have been introduced to evaluate explanation stability in these scenarios (Fel & Vigouroux, 2022). Robustness-based evaluation methods further ensure that explanations remain reliable across large perturbation spaces (Hsieh et al., 2021). Additionally, research has emphasized the importance of stable baselines and modeling uncertainty in feature attribution methods to ensure consistent and interpretable explanations when extensive input changes occur (Sturmfels et al., 2020; Slack et al., 2021). EVA (Fel et al., 2023) introduces a framework that guarantees the stability of explanations across a predefined perturbation space, ensuring that changes in unimportant features do not mislead the explanation.

## 3. Methodology

### 3.1. Background

Perturbation-based feature attribution methods assess the importance of input features by examining how changes in the inputs affect the model's output. One common approach considers that if small changes in an input feature lead to significant changes in the model's output, then that feature is important. This is often captured using gradient-based methods, where the gradient of the output with respect to the input indicates feature importance:

$$\phi_i = \frac{\partial f(\mathbf{x})}{\partial x_i}. \tag{1}$$

A larger gradient magnitude $|\phi_i|$ implies that small changes in $x_i$ can significantly affect the output. However, this small changes may not reflect the full range of input feature variations encountered in real-world scenarios.

When we increase the perturbation space by considering larger input changes, both the inputs and outputs may change significantly. To assess feature importance under these conditions, we can decompose the output changes into contributions from individual input features. By attributing the overall output change to each input feature's change, we determine feature importance based on their contributions. Methods like DeepLIFT implement this by considering both input and output differences:

$$\phi_i = (f(\mathbf{x}) - f(\mathbf{x}_{\text{ref}})) \cdot \frac{x_i - x_{i,\text{ref}}}{\sum_j (x_j - x_{j,\text{ref}})}, \tag{2}$$

where $\mathbf{x}_{\text{ref}}$ is a reference input (e.g., a baseline). This method attributes the total output change to individual input features proportionally to their deviations from the baseline. However, linearly decomposing output changes to input features is problematic because linear decomposition relies on the assumption of small perturbations, similar to a Taylor expansion.

In this paper, we consider an alternative approach: identifying features that, when significantly altered, cause minimal change in the model's output. By fixing certain parts of the input and allowing other parts to change as much as possible while keeping the output change small, we can consider the fixed parts as important features.

Considering large input changes inevitably leads to searching a vast perturbation space, which is computationally expensive, especially in high-dimensional inputs like images. To mitigate this challenge, common methods often comprise attribution precision by dividing the input into smaller regions or patches to limit the search space. For example, HSIC (Novello et al., 2022a) partitions images into blocks to assess their importance. Alternatively, some approaches sample within the perturbation space rather than exhaustively searching it (Fel et al., 2021; Scott et al., 2017; Novello et al., 2022b; Petsiuk et al., 2018). While these strategies reduce computational demands, they come at the cost of precision. Dividing the input into patches results in evaluating the importance of blocks rather than individual features, potentially missing fine-grained details. Moreover, sampling methods may not adequately explore the entire perturbation space, leading to imprecise or erroneous assessments of feature importance.

### 3.2. Our Proposed Method

Our method begins by defining the perturbation space and formulating the attribution problem as an optimization task. To address the challenge of computing mutual information within this task, we leverage SDEs to expand the perturbation space and derive a differential formulation of Fisher divergence for mutual information representation. Building on this formulation, we establish an optimization objective and framework. Finally, leveraging diffusion models, we develop an efficient method for computing Fisher divergence and demonstrate how its integral form can be used to estimate mutual information as an attribution function.

#### 3.2.1. Optimization Problems for Perturbation Space Search

We construct an optimization function that provides a direction towards the goal, significantly reducing computational costs while aiming to accurately identify important features. This approach allows us to navigate the perturbation space more effectively, enabling a practical and scalable assess-

ment of feature importance in complex models.

We begin by defining the perturbation space. The *Perturbation Space* defines how input features are modified to explore the model's response to different input variations. For a machine learning model $f : \mathbb{R}^n \to \mathbb{R}$ that maps an input feature vector $\mathbf{x} = [x_1, x_2, \ldots, x_n]^\top$ to an output $f(\mathbf{x})$, a general perturbation of a feature $x_i$ can be expressed as:

$$x_i' = \mu_i x_i + \delta_i, \tag{3}$$

where $\mu_i \in \mathbb{R}$ serves as a scaling factor to adjust the magnitude of $x_i$, and $\delta_i$ represents an additive perturbation drawn from a distribution $D_i$, introducing random fluctuations.

This formulation encompasses various perturbation strategies, including additive noise, where $\mu_i = 1$ and $\delta_i \sim D_i$ (e.g., Gaussian noise), adding random noise to the feature; small perturbations, where $\mu_i = 1$ and $\delta_i = dx$, representing infinitesimal changes in the input features, as employed by gradient-based methods such as Saliency Maps (Simonyan et al., 2014), SmoothGrad (Smilkov et al., 2017), and Integrated Gradients (Sundararajan et al., 2017); feature occlusion (masking), where $\mu_i = 0$ and $\delta_i = 0$, effectively removing or replacing the feature, with RISE (Randomized Input Sampling for Explanation) (Petsiuk et al., 2018) applying random masks to generate attribution maps; baselines, where $\mu_i = 0$ and $\delta_i = b_i$ (a baseline value), providing a reference point for comparison, as used in DeepLIFT (Shrikumar et al., 2017) with the concept of a reference input; and custom perturbations, which involve any combination of $\mu_i$ and $\delta_i$ tailored to specific requirements. The perturbed input vector is then expressed as:

$$\mathbf{x}' = [x_1', x_2', \ldots, x_n']^\top. $$

Searching the entire perturbation space to find the maximum input change with minimal output change is computationally infeasible for high-dimensional inputs. Instead, we formulate this as an optimization problem:

$$\begin{aligned} \underset{\mathbf{x}'}{\text{maximize}} \quad & \|\mathbf{x}' - \mathbf{x}\| \\ \text{subject to} \quad & |f(\mathbf{x}') - f(\mathbf{x})| \leq \xi \end{aligned} \tag{4}$$

where $\xi$ is a small threshold ensuring the output change is minimal.

While the optimization problem aims to maximize the input change $\|\mathbf{x}' - \mathbf{x}\|$ under the constraint of minimal output change, simply using the norm $\|\mathbf{x}' - \mathbf{x}\|$ as the objective function may not yield satisfactory results. This approach treats all input changes equally without considering the informational content or dependencies between features. It may overlook important structural information and fail to capture the complexity of feature interactions within the data.

To address this limitation, we adopt the mutual information approach. Considering the dataset $X$ with samples $\mathbf{x} \sim X$, we aim to optimize $\mathbf{x}'$ such that the resulting dataset $X'$ contains as little information as possible about the original dataset $X$, while ensuring that the classifier's predictions remain unchanged after perturbation. This reduction in shared information is quantified by the mutual information $I(X; X')$. Minimizing $I(X; X')$ aligns with the *Information Bottleneck* (IB) (Alemi et al., 2016) principle, which provides a theoretical framework for balancing input compression with output preservation. And the objective is defined as:

$$\mathcal{L}_{\text{IB}} = I(X; X') - \beta I(X'; Y), \tag{5}$$

where $I(X'; Y)$ is the mutual information between $X'$ and the output $Y$, ensuring that $X'$ preserves information relevant for predicting $Y$. And $\beta \geq 0$ is a trade-off parameter controlling the balance between input compression and output preservation.

### 3.2.2. PERTURBING DATA WITH AN SDE

However, directly computing mutual information $I(X; X')$ in high-dimensional data is intractable due to computational complexity (Schulz et al., 2020; Zhang et al., 2021). To overcome this challenge, we introduce SDEs to define continuous perturbation spaces. By modeling the perturbation process as a continuous transformation governed by SDEs, we can compute the mutual information changes in a continuous-time manner.

When the perturbation levels become infinitesimally small and densely packed, the cumulative effect leads to a continuous transformation of the input data. In this scenario, the noise perturbation procedure becomes a continuous-time stochastic process. To represent such a stochastic process concisely, we turn to SDEs. In general, an SDE has the following form:

$$dX_t = \mu(t)\, dt + \sigma(t)\, dW_t, \tag{6}$$

where $X_t$ represents the perturbed input at time $t$; $\mu(t)$ is the *drift term* capturing the deterministic component of the perturbation; $\sigma(t)$ is the *diffusion coefficient* controlling the magnitude of the stochastic perturbations; and $dW_t$ is the differential of a Wiener process representing the random fluctuations added to the system.

### 3.2.3. THE TIME DERIVATIVE OF MUTUAL INFORMATION

By integrating this SDE over time, we generate a continuous path of perturbed inputs $X_t$. In the following, we will demonstrate how mathematical analyses of SDEs, such as the Fokker-Planck equation, allow us to track the evolution

of mutual information over time across the entire perturbation space and compute the exact mutual information through the integration of Fisher divergence.

**Theorem 3.1.** *Let $dy_t = \mu(t)\,dt + \sigma(t)\,dW_t$ for $t \geq 0$ and $y_0 = x$. Denote by $p_t(\mathbf{y})$ and $q_t(\mathbf{y})$ the densities of $\mathbf{y}$ when the initial distribution is $p(\mathbf{x})$ or $q(\mathbf{x})$, respectively. Assume that $p_t(\mathbf{y})$ and $q_t(\mathbf{y})$ are smooth and sufficiently decaying, such that their logarithms grow at most polynomially in $t$. Then we have:*

$$\frac{d}{dt} D_{\mathrm{KL}}(p_t \| q_t) = -\frac{1}{2}\sigma(t)^2 D_F(p_t \| q_t), \quad (7)$$

*where $D_{\mathrm{KL}}$ denotes the Kullback–Leibler divergence and $D_F$ denotes the Fisher divergence.*

*Proof.* To simplify notation and improve readability, explicit references to variables (e.g., $\mathbf{x}$, $\mathbf{y}$) and integration measures (e.g., $d\mathbf{y}$) are omitted throughout the proof whenever their omission does not cause ambiguity. The proof proceeds as follows.

By applying the Lemma 1 in (Lyu, 2012) , the Fisher divergence can be expressed as:

$$D_F(p_t \| q_t) = \int p_t \big( \|\nabla \log p_t\|^2 + \|\nabla \log q_t\|^2$$
$$+ 2\Delta \log q_t \big). \quad (8)$$

Further simplifications lead to:

$$D_F(p_t \| q_t) = \int p_t \big( \|\nabla \log p_t\|^2 + \frac{\Delta q_t}{q_t}$$
$$+ \Delta \log q_t \big). \quad (9)$$

**Expanding the Time Derivative of $D_{\mathrm{KL}}(p_t \| q_t)$:**

We then begin by expanding the time derivative of the KL divergence:

$$\frac{d}{dt} D_{\mathrm{KL}}(p_t \| q_t) = \int \frac{\partial p_t}{\partial t} \log \frac{p_t}{q_t} + \int \frac{\partial p_t}{\partial t}$$
$$- \int \frac{\partial p_t}{\partial t} \log q_t. \quad (10)$$

**Elimination of the Second Term**:

The second term vanishes because the integral of $p_t$ over the entire space is constant (due to normalization):

$$\int \frac{\partial p_t}{\partial t} = \frac{\partial}{\partial t} \int p_t = \frac{\partial}{\partial t}(1) = 0. \quad (11)$$

Thus, we have:

$$\frac{d}{dt} D_{\mathrm{KL}}(p_t \| q_t) = \int \frac{\partial p_t}{\partial t} \log \frac{p_t}{q_t}$$
$$- \int \frac{\partial p_t}{\partial t} \log q_t. \quad (12)$$

Using Fokker–Planck equation:

$$\frac{\partial p_t}{\partial t} = -\nabla \cdot (\mu(t) p_t) + \frac{1}{2}\sigma(t)^2 \Delta p_t, \quad (13)$$

we decompose $\frac{d}{dt} D_{\mathrm{KL}}(p_t \| q_t)$ into two main terms:

$$\frac{d}{dt} D_{\mathrm{KL}}(p_t \| q_t) = I_1 + \frac{1}{2}\sigma(t)^2 I_2, \quad (14)$$

where the drift term $I_1$ and diffusion term $I_2$ are defined as:

$$I_1 = -\mu(t) \int \nabla p_t \log p_t + \mu(t) \int \nabla p_t \log q_t$$
$$- \int p_t \frac{\partial}{\partial t} \log q_t, \quad (15)$$

$$I_2 = \int \Delta p_t \log p_t - \int \Delta p_t \log q_t$$
$$- \int \Delta p_t \log q_t. \quad (16)$$

**Simplifying $I_1$:**

By integration by parts, $\int \nabla p_t \log p_t = 0$, so:

$$I_1 \quad = \quad -\mu(t) \int p_t \frac{\nabla q_t}{q_t} \quad - \quad \int p_t \frac{d}{dt} \log q_t. \quad (17)$$

Using the chain rule and Fokker–Planck equation:

$$\frac{d}{dt} \log q_t = -\mu(t) \frac{\nabla q_t}{q_t}, \quad (18)$$

we find:

$$I_1 = -\mu(t) \int p_t \frac{\nabla q_t}{q_t} + \mu(t) \int p_t \frac{\nabla q_t}{q_t} = 0. \quad (19)$$

**Simplifying $I_2$:** Using integration by parts:

$$\int \Delta p_t \log p_t = - \int p_t \|\nabla \log p_t\|^2, \quad (20)$$

$$\int \Delta p_t \log q_t = \int p_t \Delta \log q_t. \quad (21)$$

Thus:

$$I_2 = - \int p_t \|\nabla \log p_t\|^2 - \int p_t \Delta \log q_t - \int p_t \frac{\Delta q_t}{q_t}. \quad (22)$$

Using the definition of $D_F(p_t \| q_t)$:

$$I_2 = -D_F(p_t \| q_t). \quad (23)$$

**Combining Results**:

Since $I_1 = 0$, we have:

$$\frac{d}{dt} D_{\mathrm{KL}}(p_t \| q_t) = -\frac{\sigma(t)^2}{2} D_F(p_t \| q_t), \quad (24)$$

which completes the proof. □

**Lemma 3.2** (Time Derivative of Mutual Information). *Under the same assumptions as Theorem 3.1, let $X_t$ satisfy*

$$dX_t = \mu(t)\, dt + \sigma(t)\, dW_t, \quad X_0 \sim p(\mathbf{x}).$$

*Then the mutual information between $X_t$ and $X_0$ satisfies*

$$\frac{d}{dt} I(X_t; X_0) = -\frac{1}{2}\sigma^2(t)\, \mathbb{E}\big[\| s_t(X_t) - s_t(X_t \mid X_0)\|^2\big], \tag{25}$$

*where $s_t(\cdot)$ and $s_t(\cdot \mid X_0)$ are the score functions of the marginal and conditional distributions, respectively.*

*Proof.* By definition,

$$I(X_t; X_0) = \mathbb{E}_{p(x_0)}\big[D_{\text{KL}}\big(p_t(\cdot \mid x_0) \,\|\, p_t(\cdot)\big)\big]. \tag{26}$$

Under sufficient smoothness and decay conditions, differentiation under the integral sign is valid, giving

$$\frac{d}{dt} I(X_t; X_0) = \mathbb{E}_{p(x_0)}\Big[\frac{d}{dt} D_{\text{KL}}\big(p_t(\cdot \mid x_0) \,\|\, p_t(\cdot)\big)\Big]. \tag{27}$$

By Theorem 3.1,

$$\frac{d}{dt} D_{\text{KL}}\big(p_t(\cdot \mid x_0) \,\|\, p_t(\cdot)\big) = -\frac{1}{2}\sigma^2(t) \\ D_F\big(p_t(\cdot \mid x_0) \,\|\, p_t(\cdot)\big). \tag{28}$$

Here, the Fisher divergence can be written as

$$D_F\big(p_t(\cdot \mid x_0) \,\|\, p_t(\cdot)\big) = \mathbb{E}\big[\| s_t(X_t \mid X_0) - s_t(X_t)\|^2\big], \tag{29}$$

where the score functions $s_t(x) = \nabla_x \log p_t(x)$ and $s_t(x \mid x_0) = \nabla_x \log p_t(x \mid x_0)$ represent the gradients of the log-probability densities for the marginal and conditional distributions, respectively.

Taking the expectation over $x_0$ thus yields

$$\frac{d}{dt} I(X_t; X_0) = -\frac{1}{2}\sigma^2(t) \\ \mathbb{E}\big[\| s_t(X_t) - s_t(X_t \mid X_0)\|^2\big], \tag{30}$$

which completes the proof. $\qquad\square$

### 3.2.4. OPTIMIZATION OBJECTIVE WITH DIFFUSION MODELS

From the time derivative of mutual information:

$$\frac{d}{dt} I(X_t; X_0) = -\frac{1}{2}\sigma^2(X_t, t)\mathbb{E}\left[\|s_t(X_t) - s_t(X_t \mid X_0)\|^2\right], \tag{31}$$

we note that $\sigma^2(X_t, t) \geq 0$ and the expected squared norm is non-negative, implying $I(X_t; X_0)$ decreases monotonically

with $t$. As $t$ increases, more noise is introduced into $X_t$, reducing its mutual information with the original input $X_0$.

To leverage this property, we define a mutual information loss function proportional to $-t$:

$$L_{\text{MI}} = -t. \tag{32}$$

Our objective is to minimize mutual information while preserving the model's predictive performance. Therefore, we combine $L_{\text{MI}}$ with a task-specific loss function, such as the cross-entropy loss $L_{\text{CE}}$ for classification tasks:

$$L = L_{\text{MI}} + L_{\text{CE}}. \tag{33}$$

However, training a neural network using this loss function necessitates computing $X_t$ by integrating the stochastic differential equation (SDE):

$$X_t = X_0 + \int_0^t \mu(s)\, ds + \int_0^t \sigma(s)\, dW_s, \tag{34}$$

where $\mu(s)$ is the drift term, $\sigma(s)$ is the diffusion coefficient, and $W_s$ represents a Wiener process. Evaluating this integral at each training iteration is computationally infeasible, especially for high-dimensional data.

To address this, we refer to the **Variance Exploding (VE) Process** and the **Variance Preserving (VP) Process** (Song et al., 2020), which provide analytical expressions for $X_t$ in terms of $X_0$.

The VE process is defined by the following SDE:

$$dX_t = \sqrt{\frac{d[\sigma^2(t)]}{dt}}\, dW_t. \tag{35}$$

In practice, we choose the variance function $\sigma(t)$ as:

$$\sigma(t) = \sigma_{\min}\left(\frac{\sigma_{\max}}{\sigma_{\min}}\right)^t, \tag{36}$$

where $\sigma_{\min} = 0.01$ (*Song et al.*, 2020) and $\sigma_{\max}$ is set based on the dataset to ensure $X_1$ is approximately independent of $X_0$. Using this, we can express $X_t$ directly:

$$X_t = X_0 + \sigma(t)\epsilon, \quad \epsilon \sim \mathcal{N}(0, \mathbf{I}). \tag{37}$$

The VP process is defined by:

$$dX_t = -\frac{1}{2}\beta(t)X_t\, dt + \sqrt{\beta(t)}\, dW_t. \tag{38}$$

We typically define $\beta(t)$ as:

$$\beta(t) = \beta_{\min} + t(\beta_{\max} - \beta_{\min}), \tag{39}$$

with $\beta_{\min} = 0.1$ and $\beta_{\max} = 20$ (Song et al., 2020). Then $X_t$ can be expressed as:

$$X_t = X_0 e^{-\frac{1}{2}\int_0^t \beta(s)ds} + \sqrt{1 - e^{-\int_0^t \beta(s)ds}}\epsilon, \tag{40}$$

where $\epsilon \sim \mathcal{N}(0, \mathbf{I})$. We can explicitly compute the integral $\int_0^t \beta(s)ds$ due to the linear form of $\beta(s)$:

$$\int_0^t \beta(s)ds = \beta_{\min}t + \frac{1}{2}(\beta_{\max} - \beta_{\min})t^2. \quad (41)$$

### 3.2.5. COMPUTING THE FISHER DIVERGENCE

Computing the Fisher divergence involves evaluating the expected squared difference between the two score functions $s_t(X_t)$ and $s_t(X_t \mid X_0)$. While the conditional score function $s_t(X_t \mid X_0)$ can be computed analytically due to the Gaussian nature of $p_t(X_t \mid X_0)$, obtaining the marginal score function $s_t(X_t)$ is more challenging because the marginal distribution $p_t(X_t)$ is generally unknown and complex.

To approximate $s_t(X_t)$ for the Variance Preserving (VP) Process, we employ denoising score matching (Vincent, 2011), training a neural network $s_\theta(X_t, \sigma)$ to estimate the score function of the perturbed data distribution at different noise levels $\sigma$. Therefore, using the trained $s_\theta(X_t, \sigma)$ and the analytically tractable $s_t(X_t \mid X_0)$, we can compute the Fisher divergence between them.

For the Variance Preserving (VP) Process, we leverage the relationships derived in Denoising Diffusion Probabilistic Models (DDPM) (Ho et al., 2020), which is the discrete-time formulation of the VP process. In DDPM, the forward diffusion process gradually adds Gaussian noise to the data over $T$ timesteps, resulting in noisy samples $X_t$. These relationships allow us to express $X_t$ and $X_0$ in terms of each other and the noise components.

The forward diffusion process in DDPM is defined as:

$$X_t = \sqrt{\bar{\alpha}_t}\, X_0 + \sqrt{1 - \bar{\alpha}_t}\,\epsilon, \quad \epsilon \sim \mathcal{N}(0, \mathbf{I}), \quad (42)$$

where $\bar{\alpha}_t = \prod_{s=1}^t \alpha_s$, and the relationship between the noise scheduling parameters $\alpha_t$ and $\beta_t$ is given by:

$$\alpha_t = 1 - \beta_t. \quad (43)$$

The conditional distribution $p_t(X_t \mid X_0)$ is Gaussian with mean $\sqrt{\bar{\alpha}_t}\, X_0$ and variance $(1 - \bar{\alpha}_t)\,\mathbf{I}$. Therefore, the conditional score function can be computed analytically:

$$s_t(X_t \mid X_0) = \nabla_{X_t} \log p_t(X_t \mid X_0) = -\frac{X_t - \sqrt{\bar{\alpha}_t}\, X_0}{1 - \bar{\alpha}_t}. \quad (44)$$

To approximate the marginal score function $s_t(X_t) = \nabla_{X_t} \log p_t(X_t)$, we train a neural network $\epsilon_\theta(X_t, t)$ to predict the noise $\epsilon$ given $X_t$, as proposed in DDPM. The marginal score function can then be approximated as:

$$s_t(X_t) \approx -\frac{1}{\sqrt{1 - \bar{\alpha}_t}}\, \epsilon_\theta(X_t, t). \quad (45)$$

By substituting $X_t$ from Eq. (42) into Eq. (44), we simplify the conditional score function:

$$s_t(X_t \mid X_0) = -\frac{\sqrt{1 - \bar{\alpha}_t}\,\epsilon}{1 - \bar{\alpha}_t} = -\frac{\epsilon}{\sqrt{1 - \bar{\alpha}_t}}. \quad (46)$$

With both $s_t(X_t)$ and $s_t(X_t \mid X_0)$ expressed in terms of $\epsilon$ and $\epsilon_\theta(X_t, t)$, then we have:

$$\|s_t(X_t) - s_t(X_t \mid X_0)\|^2 \approx \frac{1}{1 - \bar{\alpha}_t}\, \|\epsilon_\theta(X_t, t) - \epsilon\|^2. \quad (47)$$

### 3.2.6. COMPUTING MUTUAL INFORMATION VIA TIME INTEGRATION

Based on our optimization framework, for each pixel value $x_i$, we can obtain a corresponding $t_i$ that represents the distance in the forward diffusion process. In the Variance Preserving (VP) process, this $t_i$ can be expressed in terms of the noise level $\sigma_i$.

For each pixel $x_i$, the mutual information $I(x_{i,t_i}; x_{i,0})$ can be similarly computed by integrating the per-pixel contribution of the time derivative:

$$I(x_{i,t_i}; x_{i,0}) = \int_0^{t_i} -\frac{1}{2}\sigma^2(t)\mathbb{E}\Bigg[ \big(s_t(x_{i,t}) \\ - s_t(x_{i,t} \mid x_{i,0})^2\Bigg] dt. \quad (48)$$

Here, $s_t(x_{i,t})$ and $s_t(x_{i,t} \mid x_{i,0})$ are the score functions for the marginal and conditional distributions of pixel $x_i$, respectively.

In practice, we perform numerical integration over discrete time steps $t_k$ to compute the mutual information:

$$I(x_{i,t_i}; x_{i,0}) \approx -\sum_{k=0}^K \frac{\beta(t_k)}{2(1 - \bar{\alpha}_{t_k})} \big(\epsilon_\theta^{\text{VP}}(x_{i,t_k}, t_k) \\ - \epsilon_k\big)^2 \Delta t. \quad (49)$$

for the VP process, and

$$I(x_{i,t_i}; x_{i,0}) \approx -\sum_{k=0}^K \frac{1}{2}\left(\frac{d\sigma_{t_k}^2}{dt}\right) \Bigg(s_\theta^{\text{VE}}(x_{i,t_k}, t_k) \\ + \frac{x_{i,t_k} - x_{i,0}}{\sigma_{t_k}^2}\Bigg)^2 \Delta t. \quad (50)$$

for the VE process, where $\Delta t$ is the time step size, and $K$ is the total number of time steps.

## 4. Experiments

In this section, we systematically evaluate our method by comparing it against several benchmark attribution approaches. For baseline methods, we selected IBA (Schulz

et al., 2020), InputIBA (Zhang et al., 2021), Integrated Gradients (Sundararajan et al., 2017), Guided-BP (Springenberg et al., 2014), DeepLIFT (Shrikumar et al., 2017), and HSIC (Novello et al., 2022a). We conduct both qualitative and quantitative analyses, with qualitative comparisons provided in the appendix. We begin with a Parameter Randomization Sanity Check experiment to ensure that our attribution method is sensitive to the learned model parameters rather than arbitrary network structures. For quantitative evaluation, we report results based on Insertion and Deletion experiments, and the Segmentation-based Ratio metric. The experimental setup are included in the appendix.

### 4.1. Parameter Randomization Sanity Check

The Parameter Randomization Sanity Check (Adebayo et al., 2018) aims to assess whether attribution methods reliably explain model behavior by analyzing their sensitivity to parameter changes. This evaluation is performed using the Structural Similarity Index Metric (SSIM (Wang et al., 2004)). A lower SSIM value between the attribution map of the original model and that of a randomized model indicates that the method is sensitive to parameter changes, effectively capturing key features influencing the model's decisions. Our experimental results demonstrate consistently low SSIM values across all layers where parameter randomization begins, highlighting the robustness of our approach in identifying essential features. We show the figure of SSIM in the Appendix.

### 4.2. Insertion and Deletion AUCs

The Deletion and Insertion methods (Zhang et al., 2021) evaluate attribution performance by progressively removing or adding pixels based on importance scores, with predictions monitored at each step to compute the Area Under the Curve (AUC). A larger difference between Insertion and Deletion AUCs (DAUC) reflects better attribution quality. As shown in Table 1, our method achieves the highest DAUC scores, outperforming baseline methods.

Table 1. Insertion and deletion experiments. The results demonstrate that our method achieves the best performance, outperforming all baselines.

| Method | DAUCs |
|---|---|
| IBA | $0.771 \pm 0.006$ |
| InputIBA | $0.833 \pm 0.002$ |
| Integrated Gradients | $0.153 \pm 0.004$ |
| Guided-BP | $0.151 \pm 0.007$ |
| Deep-Lift | $0.157 \pm 0.008$ |
| Ours | $\mathbf{0.836} \pm 0.003$ |
| HSIC | $0.133 \pm 0.002$ |

### 4.3. Quantitative Visual Evaluation via Effective Heat Ratios (EHR)

To further validate the effectiveness of our attribution method, we conduct a quantitative visual evaluation using Effective Heat Ratios (EHR). This metric assesses the concentration of attribution scores within meaningful regions, providing a structured way to compare feature importance across different methods. We perform this evaluation on the FSS-1000 dataset (Li et al., 2020), leveraging its high-quality segmentation annotations to establish ground truth regions of interest. The EHR metric quantifies the proportion of attribution scores assigned to these annotated regions, where a higher EHR indicates that an attribution method successfully localizes important features while minimizing noise in less relevant areas.

By comparing our approach with baseline attribution methods, the results in Table 2 demonstrate that our method achieves the highest Effective Heat Ratio (EHR), indicating its superior ability to focus attributions on semantically meaningful regions.

Table 2. EHR experiment results. The results demonstrate that our method achieves the highest Effective Heat Ratio (EHR), significantly outperforming all baselines. InputIBA ranks second, while other methods exhibit substantially lower EHR values.

| Method | EHR |
|---|---|
| IBA | $0.355 \pm 0.004$ |
| InputIBA | $0.466 \pm 0.005$ |
| Integrated Gradients | $0.160 \pm 0.003$ |
| Guided-BP | $0.273 \pm 0.007$ |
| Deep-Lift | $0.155 \pm 0.008$ |
| Ours | $\mathbf{0.587} \pm 0.006$ |
| HSIC | $0.121 \pm 0.003$ |

## 5. Conclusion

In this work, we introduced a novel perturbation-based feature attribution method that leverages SDEs to explore the input space in a continuous and data-adaptive manner. By formulating the process as an optimization problem and directly linking Fisher divergence to the time derivative of mutual information, our approach provides a principled theoretical framework for quantifying feature importance. Moreover, integrating the Information Bottleneck principle ensures that we identify only the most informative features without sacrificing model performance. Our empirical findings validate that this method yields more accurate and robust attributions compared to existing perturbation-based strategies, especially in high-dimensional scenarios. Moving forward, we plan to investigate the efficacy of alternative diffusion processes for even more fine-grained explanations of model behavior.

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

## A. Experimental Setup

We compared our method against several widely used baselines, including IBA, InputIBA, Integrated Gradients, Guided-BP, DeepLIFT, and HSIC. Our method is implemented using the Variance Preserving (VP) process. And the three methods (Guided-BP, Integrated Gradients, DeepLIFT) were implemented with Captum (Kokhlikyan et al., 2020) in PyTorch (Paszke et al., 2019), relying on the default parameter settings. For IBA, InputIBA, and HSIC, we ran the official code releases without altering their recommended configurations. All evaluations were conducted on a VGG16 (Simonyan & Zisserman, 2014) classification model.

## B. the Figure of Parameter Randomization Sanity Check

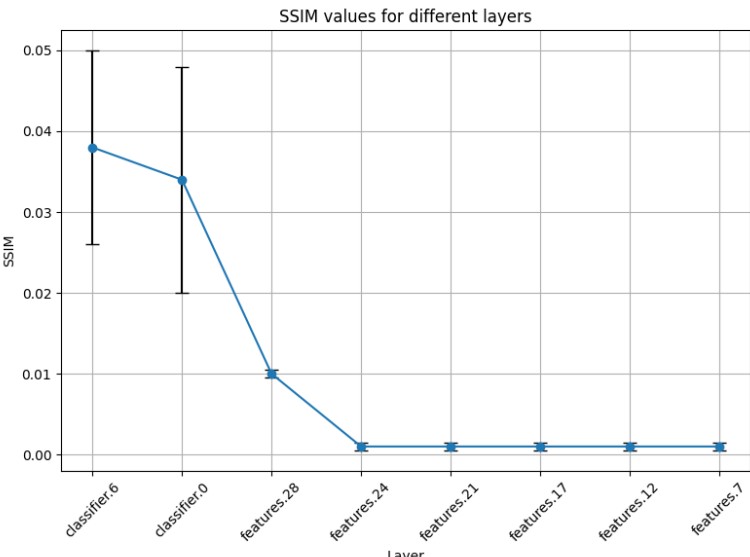

*Figure 1.* Parameter Randomization Sanity Check results. Our results show that SSIM values decrease sharply as randomization moves to earlier layers, demonstrating that our method effectively responds to parameter modifications and robustly identifies essential features.

## C. Qualitative Comparision

The qualitative results in Figure 2 reveal significant variations in the clarity and interpretability of attribution maps generated by different methods. IBA and HSIC tend to produce smooth but overly blurred visualizations, largely due to interpolation effects that obscure fine-grained details and diminish the sharpness of object boundaries. While such methods maintain a degree of consistency in attribution, they fail to capture precise feature importance at a granular level, limiting their interpretability in complex visual tasks.

In contrast, DeepSHAP and Integrated Gradients yield more detailed and fine-grained attributions, which can be beneficial for analyzing subtle model behaviors. However, their attribution maps often appear cluttered and chaotic, with dispersed importance scores that do not always align with semantically meaningful structures. This lack of spatial coherence reduces their reliability in pinpointing the most influential features driving the model's predictions.

InputIBA enhances IBA's attributions by increasing the sharpness of highlighted regions. However, it still struggles with maintaining well-defined object edges, leading to some leakage of importance scores into background areas. This dilution of feature importance can hinder precise interpretation, especially in cases where separating foreground from background is crucial.

Guided-BP stands out for its ability to generate highly detailed attributions with sharp object boundaries and intricate textures. The visual clarity of these maps makes them aesthetically appealing, but they also suffer from a major drawback: a tendency to amplify high-frequency features, which often leads to the misallocation of importance scores to irrelevant background regions. This overemphasis on fine structures can introduce misleading interpretations, as it may attribute

significance to pixels that do not directly influence the model's decision.

In contrast, our method consistently produces clear and structured attribution maps that accurately delineate object contours while avoiding unnecessary noise in the background. By balancing fine-grained feature representation with spatial coherence, our approach effectively isolates the most relevant features, providing a sharper and more intuitive visualization of the model's decision-making process. The enhanced focus and reduced attribution noise make our method particularly advantageous for applications where precise feature localization is essential.

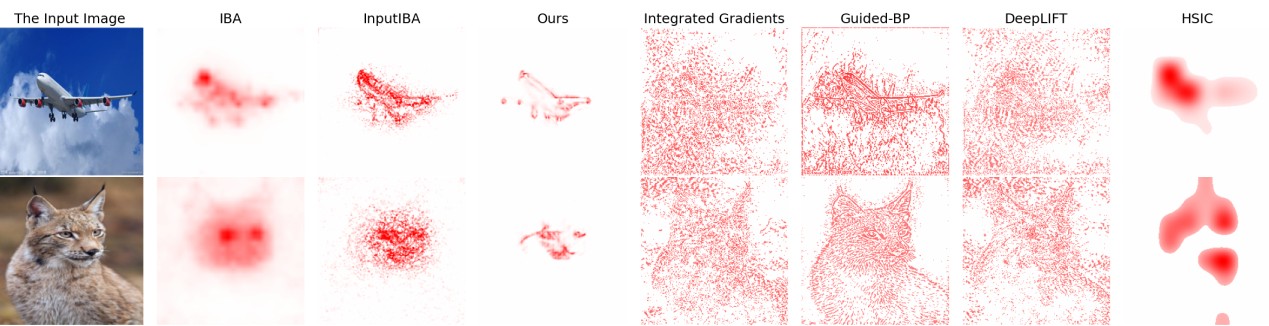

*Figure 2.* Qualitative comparison of attribution maps generated by different methods. IBA and HSIC produce smooth but overly blurred attributions, failing to capture fine-grained details. Integrated Gradients and DeepLIFT generate highly detailed attributions but appear chaotic, lacking clear spatial coherence. Guided-BP enhances edge sharpness but often misallocates importance to irrelevant background regions. InputIBA improves upon IBA but still struggles with clearly delineating object boundaries. In contrast, our method produces well-defined and focused attributions, accurately highlighting the most relevant features while minimizing noise, demonstrating superior interpretability and robustness.

