# OpenReview forum: "Fisher Divergence for Attribution through Stochastic Differential Equations"
_ICML.cc/2025/Conference — Submitted to ICML 2025_

### Official Review · Reviewer_38Uj · 2025-02-24

**Overall Recommendation:** 3

**Summary:**

The paper introduces a feature attribution framework for deep neural networks using Stochastic Differential Equations (SDEs) and Fisher Divergence. The method models continuous perturbations to explore input spaces and quantifies feature importance by linking Fisher Divergence to the time derivative of Kullback-Leibler (KL) Divergence.

The authors put forth different contributions, amongst which the development of an optimization framework based on the Information Bottleneck principle to identify informative features with minimal output change; the computation of Fisher Divergence and mutual information using diffusion models and time integration; and experiments showing how the method outperforms baselines, such as DeepLIFT, across various benchmarks..

**Claims And Evidence:**

The authors provide a good set of mathematical derivations from Fisher Divergence to the time derivative of KL Divergence for feature attribution. The few empirical results reported are also reasonable and show significant improvements over baseline methods. However, the experimental validation, in general, is quite limited, both in experimental breath and depth.

The framework’s efficiency claims lack runtime benchmarks or computational cost comparisons with simpler methods like saliency-based approaches. From what the authors present, it is unclear whether the proposed method is scalable, and can be applied to significantly larger-scale models.

**Essential References Not Discussed:**

--

**Experimental Designs Or Analyses:**

The authors do not provide extensive methodology explanations for the experiments run, although they refer to code bases in prior work for the baselines. Generally, the metrics are sound with respect to the objectives in the paper.

**Methods And Evaluation Criteria:**

The methods in the paper seem sound and tackle appropriately the feature attribution problem. The evaluations (e.g. DAUC and EHR) are reasonable and show improvements over baselines, but they are very limited in both breadth (e.g. model architecture, data distributions, data resolution etc.) and depth (e.g. model size). There is no evaluation of run-time performance or scalability of the proposed approach.

**Other Comments Or Suggestions:**

There are several minor issues such as misspellings, reverse quotes, etc.

**Other Strengths And Weaknesses:**

The paper presents some interesting findings, but the work seems quite preliminary, and not yet ready for publication. The connection between Fisher divergence and mutual information is nice, and may be a promising direction for developing more principled attribution methods.

There are, however, many unanswered questions. There is minimal discussion on the sensitivity of the method to design choices, such as the specific form of the SDE or noise schedules used in diffusion processes. The authors don't discuss potential failure cases or limitations of their framework. The experimental results are very limited (see other answers) etc.

**Questions For Authors:**

1. It is currently unclear to me how much does training a diffusion model weigh on the computational demands of the proposed method, and generally, how computationally demanding the method is with respect to varying model sizes or input dimensions. Can the authors explain?  It would be worthwhile to provide both theoretical arguments as well as empirical evidence for this.

2. Can the authors expand on the limitations of this approach? Does the method ever underperform compared to traditional attribution methods?

3. Can the author provide a higher range of experimental evidence for their claims? (e.g. varying model architecture, model size, input dimensions, data distributions etc.). It would also be helpful to extend Figure 2 with additional examples.

**Relation To Broader Scientific Literature:**

The paper builds on literature from stochastic differential equations, information theory, and diffusion models, and introduces a continuous-time perturbation framework. It also uses score-based generative models for score estimations. The novel contribution lies in the use of Fisher divergence and mutual information for more principled way to identify feature importance.

**Theoretical Claims:**

Theorem 3.1 and Lemma 3.2. both correctly apply standard results from stochastic processes, no obvious errors were spotted.

---

> ### Author Rebuttal · Authors · 2025-04-01
>
> Below we address your concerns one by one:
>
> **Experimental Designs Or Analyses:**
> Our experiments are conducted on ImageNet (3*224*224). We also provide the detail optimization algorithm in the third reviewer's rebuttal section and the score network training algorithm at the end of this rebuttal.
>
> **Questions For Authors:**
> 1. While training a diffusion model for an entire dataset can be expensive, our approach offers flexibility. We can train a diffusion model on a single image (3*224*224), which yields a score network that is sufficient for calculating pixel-wise mutual information for that image. In our experiments, generating an attribution map for one image takes about 4 minutes on a single GTX1080Ti. Moreover, when a pretrained diffusion model is available, the computation time is greatly reduced.
>   Once the diffusion model is available (whether pretrained or efficiently trained on a single image), the attribution procedure itself mainly involves evaluating the score function and performing numerical integration (e.g., for mutual information). Our method employs closed-form expressions for the Variance Exploding (VE) and Variance Preserving (VP) processes, allowing efficient computation. These operations scale linearly with the input size and are highly parallelizable on modern hardware.
>
> 2. Our approach does have limitations. If a pretrained diffusion model is not available, one must train a model on a per-image basis—which, while less demanding than training on a full dataset, still requires significant computational resources (around 4 minutes per image on a GTX1080Ti). Although our method generally outperforms traditional attribution methods such as DeepLIFT, InputIBA, and IBA on metrics (as shown in our experiments and in the second reviewer's rebuttal), there are scenarios where the additional computational overhead may not justify the performance gains compared to simpler methods.
>
> 3.  We acknowledge that our approach may be sensitive to the specific SDE form and noise schedule used. In the revised manuscript, we will include a sensitivity analysis to show how different choices of SDE parameters (such as varying $\sigma$ in Equation (36)) affect the resulting attributions. We will also discuss potential failure cases and limitations of our framework to provide a balanced perspective.
>    We appreciate the suggestion to extend Figure 2 with additional examples. We will supplement our current experimental section with further qualitative and quantitative evaluations, including more extensive visual comparisons and analyses across diverse datasets and network architectures.
>
> # Algorithm: Training Score Network for a Single Image
>
> ## Input:
> - $\boldsymbol{x}_0$: Original clean image
> - $s_\theta$: Score network with parameters $\theta$
> - $T$: Maximum diffusion time
> - $N$: Number of training iterations
>
> ## Function TrainScoreNetwork:
>
> 1. **Initialize noise schedule:**
>    - Define $\sigma_{min}$ and $\sigma_{max}$ (e.g., 0.01 and 50)
>    - Define noise schedule function $\sigma(t) = \sigma_{min} \cdot (\sigma_{max}/\sigma_{min})^t$ for $t \in [0,1]$
>
> 2. **For** $i = 1$ to $N$:
>    - Sample random time $t \sim \mathcal{U}(0, 1)$
>    - Compute noise level $\sigma_t = \sigma(t)$
>    - Sample noise $\boldsymbol{\epsilon} \sim \mathcal{N}(0, \boldsymbol{I})$
>    - Create noisy image $\boldsymbol{x}_t = \boldsymbol{x}_0 + \sigma_t \cdot \boldsymbol{\epsilon}$
>
>    - Compute true score $\nabla_{\boldsymbol{x}_t} \log p(\boldsymbol{x}_t|\boldsymbol{x}_0) = -\boldsymbol{\epsilon}/\sigma_t$
>    - Predicted score $\hat{\boldsymbol{s}}_\theta(\boldsymbol{x}_t, t) = s_\theta(\boldsymbol{x}_t, t)$
>
>    - Compute loss $\mathcal{L}(\theta) = \sigma_t^2 \cdot \|\hat{\boldsymbol{s}}_\theta(\boldsymbol{x}_t, t) - (-\boldsymbol{\epsilon}/\sigma_t)\|_2^2$
>    - Update $\theta$ using $\nabla_\theta \mathcal{L}(\theta)$
>
> 3. **Return** $s_\theta$

---

> > ### Comment · Reviewer_38Uj · 2025-04-03
> >
> > Thank you for addressing the raised concerns.
> >
> > I have raised my score, contingent on the revisions to be included (i.e. the parameter ablations, and the clarifications and limitations included in the rebuttal).

---

### Official Review · Reviewer_eU1o · 2025-03-07

**Overall Recommendation:** 2

**Summary:**

The paper considers perturbation-based methods for feature attribution. In order to employ a large perturbation space, an SDE is defined. The paper derives a connection between Fisher divergence and the KL divergence, and proposes utilizing the information bottleneck principle for optimization.The paper provides a method to calculate feature importance, with empirical validations demonstrating the method's good performances.

## update after rebuttal

The authors did provided further details on their proposed approach. However, I still have doubts concerning the validity of the surrogate loss of -t in place of mutual information. Therefore, I would like to retain my score.

**Claims And Evidence:**

The claims are supported by mathematical proofs or empirical experiments.

**Essential References Not Discussed:**

N/A

**Experimental Designs Or Analyses:**

In my view, some experimental settings are missing from the paper and even the supplements. For instance, what are the datasets used in section 4.1 and section 4.2?

**Methods And Evaluation Criteria:**

I am not familiar with the task. The empirical results do seem to suggest that the proposed method works well.

**Other Comments Or Suggestions:**

Page 4, second column, line 174 - 182: "And the objective is defined as", "And \beta >=0 is a trade-off", "And" seems to be redundant.
Page 6, second column, Equation 32: Is this correct?
Page 6, second column, line 312 (after Equation 36), "Songet al., 2020", broken reference name
Page 8, first column, line 390 - 395, "Parameter Randomization Sanity Check", "Insertion and Deletion", "Segmentation-based Ratio", misused capitalizations

**Other Strengths And Weaknesses:**

In terms of strength, using score-based approach for feature attribution is a good idea.

In terms of weaknesses, the authors' presentation is hard to follow. The proof of Theorem 3.1 takes up a lot of space, which could have been in the Appendix. Moreover, I have trouble following the proposed algorithm, as I could not find a place that explicitly give the proposed algorithm.

**Questions For Authors:**

I find the presentation of the paper rather confusing, and I would appreciate it if the authors could clarify my questions. How is the loss in Equation 33 utilized? How is mutual information used to perform feature attribution?

**Relation To Broader Scientific Literature:**

The paper is largely within the literature of perturbation based feature attribution methods. As claimed by the paper, it allows a large perturbation space.

**Theoretical Claims:**

The major theoretical result is Theorem 3.1. I briefly go through the proof of it. While I cannot determine its correctness for sure, it is quite likely to be a direct generalization of Theorem 1 in [1]. To me, the authors' presentation makes it difficult to follow what the proposed method actually is, as such I cannot verify its correctness.

[1] Interpretation and Generalization of Score Matching, Lyu, UAI 2009.

---

> ### Author Rebuttal · Authors · 2025-04-01
>
> Below, we address your specific questions and concerns:
>
> **Theoretical Claims:**
> Please see in 'Claims And Evidence' part of in the first reviewer’s rebuttal section.
>
> **Experimental Designs Or Analyses:**
> Our experiments are conducted on ImageNet, as it is a standard dataset frequently employed in prior work on attribution (e.g., IBA[Schulz et al., 2020]). For the classifier, we use VGG16, which is widely used in attribution studies [Schulz et al., 2020, Zhang et al., 2021]
>
> **Questions For Authors:**
> 1. **How is the loss in Equation (33) utilized?**
>    In our framework, the loss in Equation (33) is designed to approximate the mutual information between the perturbed input $X_t$ and the original input $X_0$. Specifically, since increasing the noise level $t$ reduces the mutual information, we use $-t$ as a proxy for the mutual information loss ($L_{\mathrm{MI}}$). This term is then combined with the standard cross-entropy loss ($L_{\mathrm{CE}}$) for classification, yielding a total loss of the form
>    $$
>    L = L_{\mathrm{MI}} + L_{\mathrm{CE}}.
>    $$
>    Minimizing this combined loss encourages the model to find perturbations that significantly reduce mutual information—thus filtering out uninformative features—while ensuring that the classifier's predictions remain correct.
>
> 2. **How is mutual information used to perform feature attribution?**
>    Our method leverages the dynamics of mutual information along a continuous-time perturbation path defined by an SDE. By linking the time derivative of the mutual information (via its connection to the Fisher divergence) to the evolution of noise in the input, we obtain a quantitative measure of how much each feature contributes to preserving the model's output. Concretely, after complete optimization, we derive a tensor $t$ (with the same dimensions as the input) where each element indicates the “time-to-noise” required to significantly reduce the mutual information of that feature. Features with lower $t$ values are deemed more important because they are less robust to noise injection—i.e., perturbing these features causes larger changes in the mutual information and ultimately in the model’s output. Moreover, using Equations (49) and (50), we can further refine these scores by integrating the mutual information dynamics over time, thus arriving at a more precise attribution map.
>
> We provide the detail algorithm for the optimaztion below for clarification.
> # Algorithm: Training $\boldsymbol{\tau}_\theta$ for Diffusion-based Feature Attribution
>
> ## Input:
> - $\boldsymbol{\tau}_\theta$: U-Net model with parameters $\theta$
> - $f$: Pre-trained classifier
> - dataset: Training dataset of images and labels
> - diffusion_type: Either "VE" or "VP"
>
> ## Function TrainDiffusionAttribution($\boldsymbol{\tau}_\theta$, $f$, dataset, diffusion_type):
>
> 1. **Initialize diffusion parameters:**
>    - If diffusion_type == "VE":
>      - $\sigma_{min} = 0.01$
>      - $\sigma_{max} =$ appropriate value for dataset
>    - If diffusion_type == "VP":
>      - $\beta_{min} = 0.1$
>      - $\beta_{max} = 20$
>
> 2. **While not converged:**
>    - Sample batch $(\boldsymbol{x}, \boldsymbol{y})$ from dataset
>    - $\boldsymbol{t} = \boldsymbol{\tau}_\theta(\boldsymbol{x})$
>    - Sample noise $\boldsymbol{\epsilon} \sim \mathcal{N}(0, \boldsymbol{I})$
>
>    - **Generate perturbed image $\boldsymbol{z}$:**
>      - If diffusion_type == "VE":
>        - $\sigma(\boldsymbol{t}) = \sigma_{min} \cdot (\sigma_{max}/\sigma_{min})^{\boldsymbol{t}}$
>        - $\boldsymbol{z} = \boldsymbol{x} + \sigma(\boldsymbol{t}) \odot \boldsymbol{\epsilon}$
>      - If diffusion_type == "VP":
>        - $\beta(\boldsymbol{t}) = \beta_{min} + \boldsymbol{t} \cdot (\beta_{max} - \beta_{min})$
>        - $\int_0^{\boldsymbol{t}} \beta(s) ds = \beta_{min} \cdot \boldsymbol{t} + \frac{1}{2} \cdot (\beta_{max} - \beta_{min}) \cdot \boldsymbol{t}^2$
>        - $\boldsymbol{z} = \boldsymbol{x} \odot e^{-\frac{1}{2} \int_0^{\boldsymbol{t}} \beta(s) ds} + \sqrt{1 - e^{-\int_0^{\boldsymbol{t}} \beta(s) ds}} \odot \boldsymbol{\epsilon}$
>
>    - $\hat{\boldsymbol{y}} = f(\boldsymbol{z})$
>    - $\mathcal{L}_{CE} = -\sum \boldsymbol{y} \log \hat{\boldsymbol{y}}$
>    - $\mathcal{L}_{MI} = \text{mean}(\boldsymbol{t})$
>    - $\mathcal{L}_{MI} = \text{mean}(\boldsymbol{t})$
>    - $\mathcal{L}$ = $\mathcal{L}_{CE} + \mathcal{L}_{MI}$
>
>    - Update $\theta$ using $\nabla_\theta \mathcal{L}$
>
> 3. **Return** $\boldsymbol{\tau}_\theta$

---

> > ### Comment · Reviewer_eU1o · 2025-04-02
> >
> > I thank the authors for engaging in discussions.
> >
> > Can the authors comment on how good a proxy -t is to the mutual information loss? I assume some approximation is employed here.
> >
> > Furthermore, in the provided algorithm, \theta is updated using \nabla_{\btheta} L, but \theta does not appear in the computational graph of L_{CE}, so it is only trained using L_{MI}. How does this relate to the algorithm provided in response to reviewer 38Uj, in which case the score network seems to be trained using standard diffusion loss?

---

> > > ### Author Response · Authors · 2025-04-07
> > >
> > > We thank the reviewer for the insightful comments and valuable feedback. Below are our responses to your questions:
> > >
> > > **Q1:** Can the authors comment on how good a proxy -t is to the mutual information loss? I assume some approximation is employed here.
> > >
> > > **A1:** Since increasing $t$ monotonically decreases the mutual information $I(X_t; X_0)$, we can use $-t$ as a proxy for the mutual information loss term, thereby avoiding the costly integral-based computation during training. Once the optimization is complete and the final $t$-tensor is obtained, we can still compute the exact mutual information using Equations (49) or (50) to produce the final attribution map. This two-stage approach—using $-t$ as a surrogate loss and then performing an exact mutual information calculation—strikes a balance between computational efficiency and theoretical rigor.
> > >
> > > **Q2:** Furthermore, in the provided algorithm, \theta is updated using \nabla_{\btheta} L, but \theta does not appear in the computational graph of L_{CE}, so it is only trained using L_{MI}.
> > >
> > > **A2:** In our algorithm, although $\theta$ does not appear directly in the expression for $L_{CE}$, the loss $L_{CE}$ is computed on the classifier output $y$ obtained from the perturbed input $z$, and $z$ is generated based on $t$, which in turn is produced by the network parameterized by $\theta$. Thus, $L_{CE}$ indirectly influences $\theta$ via the chain of dependencies $ \theta \rightarrow t \rightarrow z \rightarrow y $. Consequently, both $L_{MI}$ and $L_{CE}$ contribute gradient signals to update $\theta$ in our training process.
> > >
> > > **Q3:** How does this relate to the algorithm provided in response to reviewer 38Uj, in which case the score network seems to be trained using standard diffusion loss?
> > >
> > > **A3:** In our approach using the Variance Exploding (VE) noise addition method, we require a score network to compute the mutual information with Equations (50) (second stage as explained in A1). If a pretrained score-based diffusion model suitable for our research scenario is unavailable, we train the score network ourselves. Furthermore, when performing attribution on a small number of images, it is unnecessary to employ a diffusion model trained on a large-scale dataset like ImageNet. In our response to reviewer 38Uj, we provided an algorithm for training a score network on a single image—a method that is similar to that presented in Song et al., 2020.

---

### Official Review · Reviewer_UJF5 · 2025-03-10

**Overall Recommendation:** 3

**Summary:**

The paper studies the dynamics of the mutual information through SDE with the fisher divergence and the dynamics KL divergence. The computation process is proposed by discretization and the numerical studies apply the proposed framework in the feature attribution in the explainability of neural network.

**Claims And Evidence:**

Yes. The claim in the theoretical part in SDE and computation part of optimzation with the mutual informationis right.

**Essential References Not Discussed:**

Please compare with other work considering the perturbations about KL/Renyi divergence in SGLD like [Chourasia et al. 2022]

**Experimental Designs Or Analyses:**

Yes. I check the  Parameter Randomization Sanity Check, Deletion and Insertion methods and Quantitative Visual check,

**Methods And Evaluation Criteria:**

The proposed analysis make senses for the dynamics of mutual information in perturbation for SDE and the computation method for the loss designed for mutual information is reasonable for the implementation.

**Other Comments Or Suggestions:**

Line 498 reference needs re-organized
Line 312 the fonts are not right

**Other Strengths And Weaknesses:**

Pros:
1. The paper provides clear analysis for the mutual information dyanmics in the perturbations.

2. The paper is well-written for understanding,

Cons:
1. the technique contribution needs to be clarified. It is not news for the perturbations and the mutual information is just a direct extension.

2. restate and verify the assumptions in the paper fot he thm 3.1 and Lemma 3.2

3. the loss of the mutual information is not direct from the theoretical results

**Questions For Authors:**

1. how is the discretization error influence the mutual information integration.
2. for more complex SDE (even maybe not explicit solution), can the proposed method work
3. why ours show not signifcant improvement compared with InputIBA. I do not require this but just want to figure out the characteristics of the numerical studies.
4. what if the sigma in Eq. (36) follows a larger range.


## Update after rebuttal

Thanks for efforts of the authors. Most of my concerns are addressed but I believe the problem in the discretization error and $\sigma$ range still needs further justification. And the theoretical contribution is not so significant. Therefore, I still decide to keep my score as WA.

**Relation To Broader Scientific Literature:**

The perturbations for mutual information can be another new metric to measure the influence. The proposed can be adopted to improve the explainability of the DL model.

**Theoretical Claims:**

Yes. I check the dynamics for the KL divergence via Fish divergence, that's a mathematical manipulation.  The KL and mutual information can link the fisher divergence and score function for the computation next.

---

> ### Author Rebuttal · Authors · 2025-04-01
>
> Below are our responses to the specific concerns:
>
> **Other Strengths And Weaknesses:**
>
> 1. Technical Contribution Clarification:
>    While it is true that the mutual information dynamics under perturbations build on existing ideas, our work significantly extends these concepts by analyzing general SDEs (i.e., with time-dependent drift $\mu(t)$ and diffusion $\sigma(t)$). This extension is nontrivial and enables us to integrate the KL–Fisher divergence relationship into a new optimization framework for feature attribution—offering both theoretical insights and practical advantages in explaining neural network predictions.
>
> 2. Restating and Verifying Assumptions for Theorem 3.1 and Lemma 3.2:
>    We assume that for every $t \geq 0$, the densities $p_t(\mathbf{x})$ and $q_t(\mathbf{x})$ are twice continuously differentiable and decay sufficiently fast at infinity (i.e., there exists some $m>0$ such that $\lim_{\|\mathbf{x}\|\to\infty} \|\mathbf{x}\|^m\,p_t(\mathbf{x}) = 0$, and similarly for $q_t$). These conditions ensure that all integrals are finite, boundary terms vanish under integration by parts, and the required interchanges of differentiation and integration are justified.
>
> 3. Mutual Information Loss Proxy ($L_{\mathrm{MI}}$):
>    Recognizing that increasing $t$ (i.e., injecting more noise) monotonically reduces mutual information, we adopt $-t$ as an effective proxy for $L_{\mathrm{MI}}$. This surrogate is both computationally efficient and consistent with prior works (e.g., IBA and InputIBA), even though the theoretical derivation of the mutual information dynamics is not directly reflected in the loss formulation.
>
> **Questions For Authors:**
>
> 1. **Discretization Error in Mutual Information Integration:**
>    Once the optimization yields the tensor $t$, the integration limits for the mutual information calculation are determined, and we can choose sufficiently small step sizes for the numerical integration. This ensures that discretization errors are negligible and do not affect either the computational efficiency or the accuracy of the mutual information integration.
>
> 2. **Applicability to More Complex SDEs:**
>    Although our current experiments use SDEs with explicit solutions (e.g., the Variance Preserving or Variance Exploding processes), our framework is general. For more complex SDEs without explicit solutions, one can rely on established numerical SDE integration methods, albeit with higher computational cost, without compromising the overall approach.
>
> 3. **Comparison with InputIBA:**
> Our experiments—particularly the Effective Heat Ratios (EHR) evaluation—demonstrate that our method clearly outperforms InputIBA. We observe a significantly higher EHR score for our approach, which indicates that our attributions are more concentrated in semantically meaningful regions. Furthermore, we provide more experimental results after the 'Other Comments Or Suggestions' sector.
>
> 4. **The Range of $\sigma$ in Equation (36):**
>  Our approach is designed to explore the application of state-of-the-art diffusion models to feature attribution. At present, we employ the Variance Exploding (VE) and Variance Preserving (VP) processes, as these are the most representative and well-established diffusion models in the literature. In future work, we intend to investigate broader cases, including scenarios with larger ranges of $\sigma$ in Equation (36), to further generalize and enhance our method.
>
> **Other Comments Or Suggestions:**
>    We appreciate the reviewer’s note regarding the formatting issues. These will be corrected in the final version.
>
> **More Experiments**
> 1. Bounding Boxes Evaluation [Schulz et al., 2020]
>
> We leverage human-annotated bounding boxes from the ImageNet dataset to evaluate localization.
>
> Box-Ratio Results:
>
> | Method                 | Box-Ratio         |
> |------------------------|-------------------|
> | IBA                    | 0.997 ± 0.001     |
> | InputIBA               | 0.998 ± 0.001     |
> | Ours                   | 0.998 ± 0.000     |
> | Integrated Gradients   | 0.691 ± 0.006     |
> | Guided-BP              | 0.698 ± 0.002     |
> | Deep-Lift              | 0.695 ± 0.005     |
> | HSIC-Attribution       | 0.903 ± 0.002     |
>
>
> 2. Segmentation-based Ratio Evaluation
> Using semantic segmentation masks from the FSS-1000 dataset, we replace bounding boxes with detailed segmentation regions and compute the Segmentation-based Ratio (SR) in the same manner as the Box-Ratio.
> | Method               | SR                |
> |----------------------|-------------------|
> | IBA                  | 0.488 ± 0.004     |
> | InputIBA             | 0.468 ± 0.003     |
> | Ours                 | **0.501 ± 0.003** |
> | Integrated Gradients | 0.079 ± 0.006     |
> | Guided-BP            | 0.078 ± 0.009     |
> | Deep-Lift            | 0.080 ± 0.002     |
> | HSIC-Attribution     | 0.377 ± 0.005     |

---

### Official Review · Reviewer_SNpB · 2025-03-21

**Overall Recommendation:** 3

**Summary:**

This paper proposes a perturbation-based feature attribution method, where the input features are perturbed based on a stochastic differential equation (SDE). The proposed framework optimizes an input such that the input has small mutual information with the unperturbed input, and large mutual information with a target class. The authors theoretically show that the mutual information between the perturbed and unperturbed inputs relate to the SDE score functions, which can be approximated by diffusion models--hence computable for optimization. The mutual information with a target class is replaced with the cross entropy loss for the model to be explained. Empirical results demonstrate that the proposed method achieves better Insertion and Deletion AUC compared to baselines, as well as having better overlap with known image segmentation.

**Claims And Evidence:**

- Starting at line 107, it is claimed that a novel theoretical relationship between KL Divergence and Fisher Divergence is established. However, this result follows almost directly from Theorem 1 in Lyu et al., 2012. I wouldn't consider this as a novel contribution.

**Essential References Not Discussed:**

N/A

**Experimental Designs Or Analyses:**

- It is unclear how one would tune $\beta$ in Equation (5). It's also unclear why $\beta$ is dropped in the final loss function in Equation (33).

- The definition for $L_{MI}$ in Equation (32) is incomplete, and $L_{CE}$ is not defined. An input $x'$ should be optimized for $L$ in Equation (33) according to the conceptual framework, but on the right side of line 288 it is mentioned that a neural network is trained. The authors should clearly define $L$ and the optimization objective to prevent confusion.

- Details about the diffusion model training should be included. Is the diffusion model trained on data for which attributions are computed? This has implication for the scalability and/or generalizability of the proposed method. If a diffusion model has to be trained for each dataset, then the proposed method might not be scalable. If a pretrained diffusion model is used, then the proposed method might not transfer well to images out of distribution for the diffusion model.

- The proposed method learns a perturbed input $x'$. It is unclear how to go from $x'$ to the attribution scores.

- It's unclear what dataset was used to compute the Insertion and Deletion metrics. Also, more datasets should be included to demonstrate that the proposed approach is generalizable.

- Shapley-based feature attributions should be included as baselines, since they are considered standard feature attributions.

**Methods And Evaluation Criteria:**

The proposed method and evaluation criteria make sense.

**Other Comments Or Suggestions:**

- Much of Section 3.1 has been mentioned in Section 2. Consider merging the two sections.

- Consider moving the proofs to the Appendix.

**Other Strengths And Weaknesses:**

Overall, this paper proposes a novel approach for perturbing input features, with an optimization framework for getting perturbation-based feature attributions. However, in its current version, this paper suffers from clarity in terms of experimental details, such that a reader cannot reproduce the paper. Also, the experiments are limited in scope with respect to the number of datasets, the number of classifiers to explain, and the variety of diffusion models.

**Questions For Authors:**

N/A

**Relation To Broader Scientific Literature:**

This paper proposes a novel approach to perform perturbation for perturbation-based feature attribution.

**Theoretical Claims:**

- I checked the proofs for Theorem 3.1 and Lemma 3.2. Both look correct to me. For Theorem 3.1, the authors should mention that most of the steps follow from the proof of Theorem 1 in Lyu et al., 2012.

- What assumptions are used for $\int \nabla p_t log p_t = 0$? They should be clearly stated. Overall, after integration by parts, some terms equal to zero. The assumptions used to get those terms to zero should be stated.

- It is assumed that $p_t(y)$ and $q_t(y)$ are smooth and sufficiently decaying. These assumptions should be clearly formalized.

- The symbol $\Delta$ should be defined.

---

> ### Author Rebuttal · Authors · 2025-04-01
>
> Below, we address each of your concerns in detail.
>
> **Claims And Evidence**:
>
> Comparison with Theorem 1 in Lyu et al. (2012):
>  Theorem 1 in Lyu et al. (2012) considers the simple case of the SDE
>  $
>  \mathrm{d}Y_t = \mathrm{d}W_t,
>  $
>  where the drift is \(\mu(t)=0\) and the diffusion coefficient is \(\sigma(t)=1\). This setup corresponds to a basic Wiener process.
>
>  In contrast, our work extends this result to the much more general case of
>  $
>  \mathrm{d}X_t = \mu(t)\ \mathrm{d}t + \sigma(t)\ \mathrm{d}W_t,
>  $
>  where both the drift $\mu(t)$ and the diffusion $\sigma(t)$ are allowed to be time-dependent. This generalization is significant because:
>  1. Broader SDE Models:
>     While Lyu et al. (2012) deal with a noise model where $\mu(t)=0$ and $\sigma(t)=1$, our formulation covers a wide range of stochastic processes. This is essential for practical applications, such as those in modern diffusion models  DDPM likeand score-based diffusion models, where the noise characteristics evolve over time.
>  2. Handling of Additional Complexity:
>     Extending the proof in Lyu et al. (2012) to accommodate non-zero drift and time-varying diffusion requires new insights. For example, our proof first builds the intuition that the drift term $\mu(t)$ can be canceled out under certain conditions—a step that is unnecessary in the Wiener process setting. Similarly, the treatment of $\sigma(t)$ involves more delicate handling due to its time dependency.
>
> We follow the proof structure of Lyu et al. (2012) to enhance readability and make it easier for readers to identify which parts of our derivation are directly inherited from prior work and which parts constitute our innovations. We appreciate the reviewer’s suggestion to explicitly mention that our proof builds on the ideas of Lyu et al. (2012), and we will clarify this point in the revised manuscript.
>
> **Theoretical Claims**:
>
> 1. Please see 'Claims And Evidence' part.
> 2. - The boundary terms vanish due to suitable decay at infinity or zero boundary values for $p_t$.
>     - $\int p_t\ \nabla(\log p_t)\ dx = \int \nabla p_t\,dx = 0.$
>
> Hence,
> $
> \int \nabla p_t \log p_t dx
> \=\ 0.
> $
>
> 3. We assume that for every $t\geq 0$, the densities $p_t(\mathbf{x})$ and $q_t(\mathbf{x})$ are twice continuously differentiable and decay sufficiently fast at infinity (i.e., $\lim_{\|\mathbf{x}\|\to\infty}\|\mathbf{x}\|^m\,p_t(\mathbf{x})=0$ and similarly for $q_t$ so that all integrals are finite and boundary terms vanish.
> 4. $\Delta$ is the Laplacian operator.
>
> **Experimental Designs Or Analyses**:
> 1. We treat $\beta$ as a hyperparameter. In our experiments, we typically set $\beta = 1$, following the convention in prior works such as IBA and InputIBA. And we will clarify these details in the revised manuscript.
> 2. $L_{\mathrm{MI}}$ in our formulation leverages the observation that increasing $t$ (i.e., injecting more noise) monotonically decreases the mutual information between the original input and the perturbed input; thus, we can use $-t$ as an effective proxy for \$L_{\mathrm{MI}}$. $(L_{\mathrm{CE}}$ is the standard cross-entropy loss (please see before the Equation (33)), ensuring that the classifier’s predictions remain correct for the perturbed input. We will provided the training algorithm in the following.
> 3. We can either train a diffusion model from scratch or use a pretrained one for efficiency. If a pretrained model is used, it must be sufficiently close to the domain of interest so that its learned score function remains accurate. it is possible to train a diffusion model on a single image, and we provide the optimization training algorithm in the third reviewer's rebuttal section and score network training algorithm in the fourth reviewer’s rebuttal section due to space constraints.
> 4. After the complete optimization, we obtain a tensor $t$ with the same dimensions as the input image; this tensor effectively encodes the "time-to-noise" for each pixel, which we can interpret as a preliminary measure of feature importance—pixels with lower $t$ values indicate features that are more critical for preserving the model’s output. Moreover, by further computing the integrated mutual information along the diffusion path using Equations (49) and (50), we can yield a more precise attribution map.
> 5. Currently, our experiments are conducted on ImageNet, as it is a standard dataset frequently employed in prior work on attribution (e.g., IBA[Schulz et al., 2020]). For the classifier, we use VGG16, which is widely used in attribution studies [Schulz et al., 2020, Zhang et al., 2021] and provides a reliable benchmark. Moreover, incorporating additional classifiers is possible, and we plan to include them and show the comparison between our methods and others in revised version.
> 6. We will compare our method to at least one Shapley-value-based approach, acknowledging that Shapley methods are standard baselines in feature attribution.

---

> > ### Comment · Reviewer_SNpB · 2025-04-08
> >
> > The authors' response has addressed my clarification questions. I encourage the authors to include those clarifications. In the paper, it should be clearly indicated when a pretrained diffusion model is used and when a diffusion model is trained for each image.
> >
> > I have accordingly raised my score. However, the updated score is contingent on the following.
> > - Clear details about experimental settings in the paper.
> > - More empirical results with additional classifiers, datasets, and (pretrained) diffusion models.
> > - Inclusion of Shapley-based attribution methods.

---

### Decision · Program_Chairs · 2025-05-01

**Decision:**

Reject

**Comment:**

This paper introduces a novel perturbation-based feature attribution method that leverages stochastic differential equations (SDE) to model continuous perturbations and comprehensively explore the perturbation space. The proposed method employs SDE to define the perturbation space that is more flexible than that in most existing perturbation-based methods, and develops an optimization framework for feature attribution avoiding exhaustive search in the entire perturbation space. The proposed method is theoretically motivated and empirically evaluated.

Overall, most reviews are positive for this submission, although noting issues such as lack of clarity in some technical content, limited significance in the theoretical contribution, and insufficient empirical evaluation. The authors are encouraged to fully leverage the reviewer feedback to further improve the paper.